# Combined Particle Swarm Optimization and Reinforcement Learning for Water Level Control in a Reservoir

**DOI:** 10.3390/s25165055

**Published:** 2025-08-14

**Authors:** Oana Niculescu-Faida, Catalin Popescu

**Affiliations:** Hydrotechnical Engineering Department, System Engineering—Automation and Applied Informatics Domain, Faculty of Hydrotechnics, Technical University of Civil Engineering Bucharest, Blvd. Lake Tei No. 122-124, Sector 2, 020396 Bucharest, Romania; catalin.popescu@utcb.ro

**Keywords:** control system, reinforcement learning, particle swarm optimization

## Abstract

This article focuses on the research and advancement of an optimal system for the automatic regulation of the water level in a reservoir to eliminate flooding in the area where it is located. For example, in this article, the regulation of the level in the Mariselu Reservoir from the dam in Bistrita–Nasaud County, Romania, was considered as a practical application. Industrial PID controller tuning provides robust and stable solutions; however, the controller parameters may require frequent tuning owing to uncertainties and changes in operating conditions. Considering this inconvenience, an adaptive adjustment of the PID controller parameters is necessary, combining various parameter optimization methods, namely reinforcement learning and Particle Swarm Optimization. A new optimization method was developed that uses a mathematical equation to guide the Particle Swarm Optimization method, which in essence enhances the fitness function of reinforcement learning, thus obtaining a control system that combines the advantages of the two methods and minimizes their disadvantages. The method was tested by simulation using MATLAB and Python, obtaining very good results, after which it was implemented, which successfully prevented floods in the area where it was placed. This optimal automation system for dams should be implemented and adapted for several dams in Romania

## 1. Introduction

The problem of flooding is of great international importance, as it has major consequences for the environment, economy, public health, and sustainable development. It represents one of the most frequent and destructive forms of natural disaster and is amplified by climate change, rapid urbanization, and environmental degradation.

### 1.1. Impact of Floods at International Level

The problem of flooding is of great international importance, as it has major consequences for the environment, economy, public health, and sustainable development.

International flood impacts:Floods cause thousands of deaths annually, forcing millions of people from their homes. Regions with poorly developed infrastructure, such as Southeast Asia and sub-Saharan Africa, are particularly vulnerable.Infrastructure destruction:Roads, bridges, buildings and other essential structures are frequently destroyed, which can paralyze transportation, trade, and public services. The reconstruction costs are often high.Economic impact:Flooding affects agriculture through the loss of crops and fertile land, which directly affects food security. It also affects industrial activities and tourism, reducing national GDP in many countries.Public health issues:Flooding can contribute to the spread of communicable diseases, such as malaria, dengue fever, and cholera, due to contaminated water and lack of hygiene. However, access to drinking water and food is a major challenge.Environmental degradation:Natural ecosystems are affected, and wildlife habitats can be destroyed. Waste and chemical pollution during floods have long-lasting effects on the environment.Role of climate change:Rising sea levels and extreme weather events, such as torrential rain, have made flooding more frequent and severe. This particularly affects coastal communities and cities along large rivers.

### 1.2. The Role of Dams in Flood Prevention

Dams play an important role in flood protection, which is one of the most dangerous natural disasters, as they can lead to enormous loss of life and economic losses worldwide. Unfortunately, many studies predict a worsening in the future, due to the growth and extent of these phenomena [1]. Many factors drive this trend, among which, climate change [2] and changes in land use are the most important. In response to these worrying statistics, dams play a crucial role. In the context of climate change, hydropower generation is expected to increase significantly in the future owing to the increasing demand for renewable energy, although climate change may affect river flow and, consequently, the availability of water for power generation [3].

### 1.3. Related Literature on Flow Monitoring and Prediction

In [4], the authors propose a method to solve the problem of maintaining optimal water flow in the Zayandehroud dam. In this sense, they proposed a method for regulating water flow in a dam based on a prediction model. The experimental results show that the method can accurately predict changes in water flow and effectively regulate dams to maintain optimal flow, which provides an important reference for lake management and protection.

A model based on prediction was also used by the authors of [5], who, for the selection of the optimal model for the purpose of predicting the flow of two dams, generated sixteen scenarios considering two dams, two climatic conditions, and four deep learning models.

While such methods improve forecasting accuracy, they often lack an active control loop that continuously adapts actions based on real-time system states.

A similar approach can be considered for the application of machine learning to predict dam behavior combined with the detection of data anomalies. Focus was given to the practical application of advanced predictive models within a dam monitoring system, enabling the detection of anomalies at an early stage and improving maintenance and failure prevention [6].

Automatic learning techniques are also used in the study [7], in which their effectiveness to predict flow, namely multivariate linear regression and three artificial neural networks: multilayer perceptron, nonlinear exogenous autoregressive, and long-term memory.

The primary models frequently analyzed in dam-related research include neural networks, Support Vector Machines (SVMs), and boosted regression trees [8].

However, the models used in [6,7,8] function primarily as predictors rather than decision-makers and are not inherently suitable for real-time control adaptation.

Another important aspect is dam safety, the subject being treated by the authors of another study [9], where the principles, shortcomings, and development trends of three types of data analysis monitoring methods regarding dam safety monitoring are synthesized. The dam monitoring data analysis methods studied in this study are the environmental variable monitoring model, monitoring index, and abnormal value detection methods for dam conditions.

In [10], the authors referred to data monitoring for the safety control of the discharge flow from a dam. A learning machine was trained to predict dam leakage, where the parameters were adaptively optimized using the Jaya algorithm. The simulation results of the case study showed that a satisfactory prediction of runoff flow was achieved.

These approaches, although promising for anomaly detection or flow estimation, do not close the loop for control action generation, nor do they employ learning agents capable of autonomous policy improvement.

The aim of the research [11] is to forecast the inflow of the Dez dam reservoir by using auto regressive moving average and auto regressive integrated moving average models while increasing the number of parameters in order to increase the forecast accuracy to four parameters and compare them with static and dynamic artificial neural networks. By comparing the root mean square error and mean bias error, the dynamic artificial neural network model with sigmoid activity function was chosen as the best model for forecasting Dez reservoir inflow. The dam accumulation flow for the last 12 months shows that the auto regressive integrated moving average model had a smaller error compared to the auto regressive moving average model.

Yet, these are passive models and offer no mechanism for real-time policy optimization under dynamic environmental conditions.

### 1.4. A Comparison Between Different Similar Control Methods and the Proposed PSO + RL Method

We compared some reward learning methods studied in this paper with other similar studies. For example, “Optimal output tracking control with model-free feedback for two-dimensional batch processes” and “Optimal adaptive output tuning based on output feedback for nonlinear systems with strict continuous-time feedback”. For the two methods, we studied the advantages and disadvantages, we compared each method with the proposed PSO + RL method presented in the paper, and then we concluded which of the methods would be more suitable for the studied application.

#### 1.4.1. Output Feedback Based Adaptive Optimal Output Regulation for Continuous-Time Strict-Feedback Nonlinear Systems

This method is used if you have a well-understood system with a clear structure and you want to achieve robust adaptive control in real time.

This is a sophisticated control technique that brings significant advantages in regulating the level in a reservoir, especially when the system is uncertain and not completely observable (i.e., you do not have access to all the internal variables of the system, but only to some outputs—such as the water level).

Advantages:Does not require measurement of internal states—relies only on observable outputs.Adaptive—the lake is a system with high uncertainties: meteorological variations, tributary flows, evaporation. This method adapts in real time to changes, even if the exact model of the system is not completely knownOptimal—the method optimizes performance against a reference, not just stabilization. It does not just maintain the level within a range, but makes it follow an optimal trajectory (maintaining an optimal average level for flood safety)Suitable for strict-feedback systems—the output depends on a series of subsystems linked in a chain.Guaranteed stability (Lyapunov-based)—the controller is designed to be mathematically stable, using Lyapunov theory. This means you can guarantee that the water level will not become uncontrollable, even under perturbation conditions.

Disadvantages:
Requires fine and sensitive tuning of the parameters. Adaptation parameters, feedback gains, and Lyapunov functions must be tuned carefully. If you choose inappropriate values, you may experience oscillations, slow behavior, or even temporary instability.Higher complexity than classical control (e.g., PID).Requires an adaptive observer in full versions—may increase computational cost.Possible slow convergence. Because it is adaptive, the method has a “learning” or adjustment phase. In applications with rapid variations (e.g., sudden torrential rain), it may adapt too slowly to react effectively in time.

##### Comparison Between Output Feedback Adaptive and PSO + RL Methods

In Table 1, there is comparison between the Output Feedback Adaptive and PSO + RL methods.

##### Conclusion

If an intelligent, robust and optimal control system for a reservoir is desired, and the exact model is not completely known, this method is highly suitable. It is superior to a PID control, especially in the presence of external uncertainties and variability.

If the project is critical (dam, population protection), it is preferable to start with this method for safety and stability, if the system dynamics can be expressed as a sequence of differential equations, where each state depends on the previous state and the control. However, the system response when applying this method is much slower than the system response using the PSO + RL method.

#### 1.4.2. Model-Free Output Feedback Optimal Tracking Control for Two-Dimensional Batch Processes

This method is well suited for complex, repetitive industrial processes that undergo frequent variations, such as systems with unforeseen or unknown disturbances.

The method is a classic case of level control in a reservoir approached with a data-driven learning approach, without an explicit model.

The context of the problem is the following:The input is H(t) the actual water level measured by the sensor;The output is Dv(t) the gate opening;The gate control is desired so that the water level H(t) follows a desired value Hd(t).

The objective of the method is to use an optimal strategy for opening the gate Dv(t) so that the level H(t) follows a reference Hd(t), using a reinforcement learning (RL) algorithm adapted to the process, based on system data (without using an explicit mathematical model).

##### Advantages and Disadvantages of the Method

This method is well suited for complex, repetitive industrial processes that undergo frequent variations, such as systems with unforeseen or unknown disturbances.

The advantages of the method are as follows:No need for an explicit mathematical model: There is no need to identify the differential equations that describe the process. The system learns only from the data. It is the ideal method for complex or poorly modeled processes.Based on real-time monitored data: The method learns from input-output data collected in real time.Feedback only on output: It does not require measuring all internal states of the system but only the observable outputs.Optimal tracking: It allows optimal control over a desired value (e.g., maintaining the level), not just stable regulation.Suitable for 2D processes: Extend the method to systems with two dimensions; for example, time and level.Flexible and adaptable to changes: It automatically adapts if conditions change (e.g., lake level rises due to rain). Other perturbations (precipitation) can be added to the state data without changing the fundamentals of the method.

The disadvantages of the method are as follows:Requires large amounts of data: The quality of control depends on the volume and diversity of historical data available. If the data is incomplete, performance de-creases.Computational complexity: Learning algorithms (especially in 2D and with Q functions) can require large resources (memory, processing time), especially in real-time applications.No formal guarantees of stability: In many cases, control stability cannot be guaranteed theoretically, especially in online learning. Validation through simulation or experimental testing is required.Difficulty in tuning: The choice of RL parameters (learning rate, exploration policy, discount factor) strongly influences performance and may require expertise.Risk of overlearning: If historical data does not cover enough variation, the learned policy may perform poorly in new conditions.Low interpretability: Compared to classical control (PID), the learned policy is often difficult to understand or explain to human operators.

##### Control Method Comparison Between This Method and the Method Presented in the Paper

Both are methods based on machine learning but have different concepts and uses. Below is a detailed and clearly structured comparison.

In Table 2, the comparative advantages of the two methods are outlined.

Table 3 presents the comparative disadvantages of the two methods.

##### Conclusions

In the case of the application studied in the paper, the system has large variations, so the PSO + RL method is more suitable due to the following characteristics:It learns control policies optimally from interaction with the environment;It can adapt the control strategy to unpredictable conditions (drought, torrential rains);Optimization with PSO can improve the parameters of the RL network for increased performance.

Model-free output feedback optimal tracking control is more suitable for a simple system for the following reasons:It does not require the system model;It is directly applicable to the output feedback;It is stable, easier to implement, and can work well in less variable environments.

### 1.5. The Need to Optimize Control with Intelligent Algorithms

In this context, to mitigate the effects of climate change and increase energy efficiency, it is crucial to develop advanced control systems to optimize processes. Automatic industrial processes require a proportional–integral–derivative (PID) controller to obtain the output signal of the set value [12]. PID controllers are efficient if the system does not change its initial conditions [13]. However, if this is the case, the controller must be re-adjusted, which affects the response times [14]. Artificial intelligence techniques, especially reinforcement learning algorithms, offer the option of automatically and continuously adjusting controllers based on real-time feedback from the system [15].

While literature contains studies that individually apply reinforcement learning [16,17] or Particle Swarm Optimization [18] to water systems, their hybrid use is still rare and mostly limited to tuning PID controllers or offline parameter optimization. These works do not address the critical integration of PSO with RL for online policy enhancement, nor do they leverage domain-specific physical models to guide optimization.

The present work focuses on PID regulation and hybrid PSO-RL methods but there are control strategies for reservoirs used to store excess energy that emphasize multi-objective formulations. These approaches integrate economic signals (such as real-time energy prices), environmental constraints (e.g., ecological flow limits), and institutional regulations in the control design. These frameworks move beyond classical level regulation and provide adaptive, policy-compliant, and economically sustainable solutions [19]. Although the main purpose of the proposed control system is to prevent ecological and human disasters, including economical constraints, when the situation allows it, would be a significant step forward. Future work should explore integrating these strategies with intelligent control agents to ensure robust, real-time dam management in the context of climate variability and socio-economic pressure.

### 1.6. Practical Application of the Paper

This paper studied the level regulation of the Mărișelu reservoir at the dam in Bistrița, Năsăud County, Romania. The Mărișelu Dam is part of the Someșul Mic Hydropower System and plays an important role in generating electricity, regulating the river flow and preventing floods.

Heavy rainfall and flash floods, as well as inadequate management of the volume of water stored and released, have necessitated the development of an optimal reservoir control system to prevent flooding. Water volume management involves preemptive water level control, i.e., lowering the reservoir level before a major storm to create additional capacity.

PID controller parameters may require frequent adjustments owing to frequent changes in environmental conditions that influence operating conditions.

Therefore, a new optimization method has been developed that uses a mathematical equation to guide Particle Swarm Optimization, which in turn optimizes the fitness function of reinforcement learning, thus achieving a control system that quickly adapts to the conditions of the environment and acts early to prevent floods in that area.

The method was tested by simulation using MATLAB 9.10 and Python 3.6.15, obtaining very good results, after which it was implemented, which successfully prevented floods in the area where it was placed.

PSO and RL have advantages and disadvantages that make them suitable for certain types of applications.

Particle Swarm Optimization is a simple algorithm, requires little computing power, and is very fast [20]. The disadvantages of this method are that it can get stuck in local optimal solutions without reaching an optimal solution over the entire range and that it is not adaptable to environmental changes [21].

Reinforcement learning is not very fast, but it offers an optimal solution throughout the entire study interval and is adaptable to influences from the external environment [22]. To operate efficiently, this system requires a large database to train the search agent.

After achieving satisfactory training performance on the model, the agent was tuned online in the actual process to adapt to real dynamics. The simulations demonstrated the successful use of machine learning algorithms to adjust the parameters of the PID controller for optimal control of the reservoir-level regulation system.

### 1.7. Original Contributions of the Study

#### 1.7.1. The Difference Between What Exists and the Study Presented in the Paper

The study presented in the paper is differentiated as follows:It proposes a hybrid combination of PSO and RL, in which PSO is used as a pre-training or initial optimization method for RL, which is not frequently encountered in the dam control literature or in water control applications;Integration with real SCADA systems and physical testing, not just in simulation, provides added practical value.

#### 1.7.2. The Difference Between the Bibliographic References and the Study Presented in the Paper

References from the literature that are comparable to the study presented in the paper include the following:Ref. [16] use RL for water level control, but do not integrate PSO;Ref. [18] optimize a PID with PSO for flow control, but without real-time adaptation through RL;In the work of [17], an intelligent control model with RL for hydropower plants is developed, but the lack of physical testing limits its applicability.

#### 1.7.3. The Main Difficulty of the Proposed Method

The central difficulty of the method applied in this paper lies in the coherence and stability of the combined control between two very different methods:PSO, which searches globally and offline;RL, which learns locally, online, and can have unstable behaviors if not well tuned.

#### 1.7.4. Clarity of Contribution

This paper proposes an innovative hybrid method that combines global optimization through PSO with local adaptability provided by RL for automatic water level regulation in a reservoir.

In contrast to existing studies, which approach these methods separately or only in simulation, the proposed solution is tested both in simulation and in a physical system, being integrated into a real SCADA architecture. This approach highlights the potential of combining optimization algorithms and adaptive learning in a critical control system, providing superior performance in dynamic and unpredictable environments.

Unlike traditional hybrid approaches where Particle Swarm Optimization (PSO) is used merely for hyperparameter tuning, our method directly optimizes the policy or value function of the reinforcement learning agent, using PSO guided by a simplified nonlinear mathematical model of the reservoir system. This restricts the solution space to physically meaningful regions and accelerates convergence to practical control policies. Moreover, this approach ensures that exploration in early learning stages avoids unsafe system states, an advantage rarely addressed in prior PSO-RL implementations. The integration of this method into a real SCADA-controlled dam infrastructure further demonstrates its applicability beyond simulated environments.

### 1.8. Real-World Applications

Automated dam control has a significant impact in many critical areas:Flood prevention in urban areas:

Examples such as the Thames Barrier in London or the flood protection system in the Netherlands (Delta Works) use automated control systems to prevent major damage caused by rising water levels.

Efficient hydroelectric power generation:

Hydroelectric power plants in Norway, Canada or Romania (e.g., Vidraru, Portile de Fier) use automated systems to regulate water flow, optimizing energy production according to grid demand.

Smart agricultural irrigation:

In India or Israel, dams are managed with automation to efficiently distribute water to irrigation systems, reducing losses and waste in an increasingly dry climate.

SCADA systems for smart dams:

Modern dams, such as the Hoover Dam (USA) or Itaipú (Brazil–Paraguay), integrate SCADA with machine learning algorithms to detect abnormal behavior in real time, preventing costly failures.

## 2. Automation Application

To achieve the automatic level regulation system in a reservoir, an industrial monitoring, command, and control solution was chosen consisting of a system made up of a PLC programmable controller (Schneider Electric TM221CE16R, Le Vaudreuil, France), HMI human-machine interface (Delta DOP-107EG, Taoyuan City, Taiwan), CM electrical parameter measurement center (Schneider Electric A9MEM3150, Le Vaudreuil, France), GPRS router (Teltonika RUT200, Kaunas, Lithuania), and two SCADA units, fixed (procesor Intel^®^ Core^TM^ i5-6200U,16GB DDR4,480GB SSD, Intel HD Graphics 620, China) and mobile (iHUNT Titan P10000 ULTRA, China).

The realization of the automation application was possible through a project with code UTCB-CDI-2022-011, won through a national competition and financed by the Technical University of Construction Bucharest, Romania.

### 2.1. The Electrical Project

The electrical project of the automation system included the design and sizing of the control and power circuits, considering the following:Power supply of automation equipment (PLC, sensors, actuators);Electrical protection through automatic fuses and contactors;Integration of input/output modules for data acquisition and device control;Interconnection with the SCADA system via Ethernet/serial network.

In Figure 1, the general supply part is a four-pole fuse, Q0, which supplies the entire panel passing through the measuring unit, CM. The rest of the equipment is powered through the three bipolar fuses: Q1, Q2, and Q3.

In Figure 1, the first fusion is for the socket that powers the router. The second fuse is used for source V, which outputs a 24 volts direct current that feeds the switch, the HMI, and is also used on the control side (to the keys, PLC) and on the signaling side to the PLC and to the lamps. The third fuse was used to supply PLC at 220 V.

The automation system was equipped with a programmable controller, network parameter analyzer, human–machine interface (HMI).

### 2.2. The Components Inside the Automation Panel

After developing the electrical project for the automation system, the electrical/electronic components were integrated into a metal panel with dimensions of 500 × 400 × 200 mm (Figure 2). The electrical panel was machined and labeled with respect to the electrical diagram. Functional tests proved that the automation panel met all requirements.

The automation table in Figure 2 contains the following components:

Q0—four-pole fuse;

CM—measurement center (network analyzer).

The electrical panel has an HMI, two selection keys, and three signal lamps in one RGB.

The first selection key is manual-0-automatic.

The second selection key is conditioned by the first; the second key, which is used for closing or opening, works only if the first key is in manual mode.

There are three lamps in each RGB: red, green, and blue.

Green appears when the gate is in the process of opening, and blue when the gate is commanded to close.

### 2.3. SCADA Architecture

In Figure 3, there is a diagram of the SCADA communication architecture in which all the equipment and the connections between them appear.

As shown in Figure 3, the programmable controller communicates with the network analyzer or the CM through the MODBAS RTU RS-485 serial interface.

The PLC, router, and HMI enter a switch and communicate through a MODBUS TCP/IP interface, via an Internet cable. The local information arrives from the PLC, through a switch in the HMI.

For SCADA, which is found on the laptop, the connection is made with the electrical panel via Wi-Fi to the router, just as the connection is made between the telephone and the electrical panel.

The sensors integrated in the SCADA system for regulating the water level in the reservoir where the dam is impounded are as follows:Apure KS-SMY2 Water Hydrostatic Level Transmitter (GL Environmental Technology Co., Ltd., Shanghai, China), 4–20 mA, 0–10 V.Opening and closing the segment gate with the flap are performed by means of left and right chain mechanisms. Each mechanism is driven by a motor controlled by a frequency converter.

The discharged flow is calculated by interpolation depending on the upstream level and the gate opening.

As shown in Figure 4, level control is desired with a delay of 10 s, until the controller part begins to operate, increasing the opening of the sluice, so that the actual level (LEVEL [m]), which is now 70.25, drops to the set value of level (SETPOINT level [m]) 70.1.

In Figure 4, the dam will start to open and as more water is released, the level drops. The actual measured level (LEVEL [m]) gradually decreases by 0.1 until it reaches the set level value (SETPOINT Level [m]). The controller works as long as needed, until the actual level (LEVEL [m]) reaches the set level (SETPOINT Level [m]). At the last opening of the sluice, the actual level (LEVEL [m]) reached the value of the set level [m]. The sluice was opened gradually to avoid wear.

## 3. Making the Software for the Automation Application

Section 3 describes the creation of the programmable automatic software, local control and visualization interface software (DOPSoft 4.00.10) for HMI, and the monitoring, control, and data acquisition system software (TeslaSCADA_Runtime) for SCADA.

### 3.1. Realization of the Software for the Programmable Automaton

To create the code for the programmable automaton TM221CE16R, the EcoStruxure Machine Expert-Basic V1.2 SP1 programming environment was used.

The level adjustment subroutine performs level adjustment in the reservoir using a PID-type controller. The output of the controller changes the opening of the sluice in real time depending on the need, so that the level is maintained at the reference level. Table 4 defines the values for the PID controller parameters: proportionality constant Kp, integration time Ti, derivation time Td, and minimum/maximum regulation.

There was a delay that caused the controller to start after 10 s.

The sluice subroutine performs local sluice commands in the local or remote mode, manually or automatically (Figure 5). Depending on the actual opening of the sluice and the command imposed by the programmable automaton or the selection key on the automation panel, the sluice receives the command to open or close, and the signals of these commands are shown in the local lamp signaling.

### 3.2. Realization of Monitoring, Control, and Data Acquisition System Software (SCADA)

The “TESLA SCADA” programming environment was used to create the SCADA software.

Data are taken from the PLC every second, from what the SCADA system has acquired for a year, and the values of the parameters—level, sluice opening, and discharged flow—which were saved in a database created by us for the history of values.

The three pieces of information that interest us have been defined in the database, so that we can display on the map in Figure 6 the evolution over time of the three parameters: level, sluice opening, and discharged flow.

Figure 7, below, describes the link between the different components of the system.

The AI agent runs on a local server connected to SCADA via OPC UA, being responsible for generating the optimal command for the gate position every second. The PSO + RL controller adjusts the commands sent to the TM221CE16R PLC, which executes the physical actions. The AI module does not run directly in the PLC, but in parallel, being continuously monitored. In case of communication error or unacceptable performance, the system automatically switches to a classic PID control version configured in SCADA.

## 4. Optimization of PID Controller Parameters

Industrially, the choice of PID parameters is based on experience, and is achieved through trials [23]. Industrial PID controller tuning provides robust and stable solutions, but the controller parameters may require frequent tuning owing to uncertainties and changes in the operating conditions [24]. For this reason, there is a need for optimization methods of PID controller parameters that can adapt to the environmental conditions to mitigate the risk of flooding in dams.

In this section, two approaches for optimizing the parameters of the PID controller were tested using MATLAB and Python, highlighting the advantages and disadvantages of these methods: reinforcement learning and Particle Swarm Optimization.

### The Need to Optimize PID Parameters

A PID controller works optimally only when its parameters are well adapted to the system characteristics. However, in the case of dams the following elements present:Weather conditions (precipitation, snowmelt) can vary suddenly;The inlet flow is difficult to control;Reaction processes (opening of valves) have delays and nonlinearities.

In this context, traditional control (manual or fixed methods) is not sufficient. Therefore, dynamic optimization methods are proposed, with machine learning algorithms.

## 5. The Proposed Method of Optimizing the Parameters

Combining the advantages of reinforcement learning and Particle Swarm Optimization results in a tuning system with an improved response speed. The PSO method lacks a reference point at the start of the search for the optimal solution.

In contrast to traditional PSO, which explores the solution space without accounting for system dynamics, the proposed hybrid approach incorporates a nonlinear differential equation that models the reservoir’s water level evolution. This physical model acts as a constraint and guide during the optimization process, enabling the PSO algorithm to evaluate particle fitness not just heuristically, but based on predicted behavior consistent with fluid dynamics by redefining the objective function. This reduces the search space, improves convergence speed, and ensures that the solutions proposed by PSO remain physically viable and safety-compliant.

Next, we will apply Particle Swarm Optimization to improve reinforcement learning policies or value functions, with the mathematical model providing estimates to direct the particles to the best solutions.

To connect Particle Swarm Optimization (PSO) with reinforcement learning (RL) in a Python program, a system was created that uses PSO to optimize parameters or strategies in an RL agent.

### 5.1. Comparison Between the Optimization Methods of the PID Controller

In Table 5, a general characteristics comparison is made between the optimization methods of the tested PID controller parameters: reinforcement learning and Particle Swarm Optimization.

From the analysis of the advantages, disadvantages, and performances obtained with the optimization methods tested in this study, it was found that the reinforcement learning adjustment method is the only adaptive method to external factors, which recommends it as the main adjustment method.

In addition, we chose this method to optimize the controller parameters because reinforcement learning learns directly from real-world data and efficiently manages complex and uncertain scenarios.

To achieve high performance, the agent must be trained well. This requires a large volume of data and a considerable computing capacity. To reduce this inconvenience, another regulation system explores the solution space at a much higher speed than reinforcement learning. For this task, the Particle Swarm Optimization tuning type is the optimal solution.

The disadvantage of Particle Swarm Optimization, when used alone, is that it can converge to local solutions. However, if it is used to guide reinforcement learning to the optimal solution, this disadvantage disappears because the agent explores the entire solution space.

### 5.2. Optimal Tuning of PID Controller Using Particle Swarm Optimization

The number of particles in a Particle Swarm Optimization (PSO) algorithm plays a crucial role in finding possible solutions in the search space [26]. A larger number of particles leads to a deeper exploration of the search space, which can increase the chances of finding an optimal solution [27].

The equation for updating the velocity is a key factor that governs the movement of particles within the search space [28].

The initial step in analyzing the movement of particles within a swarm is to establish their starting position within the search space. This position is usually represented by a vector of real-valued numbers and is used to encode the values of the variables being optimized.

A higher cognitive learning factor makes the particle more influenced by its own past performance, whereas a lower cognitive learning factor makes the particle more responsive to changes in the search space [29].

The swarm topology is a key aspect of Particle Swarm Optimization, as it defines the manner in which particles are interconnected and interact with one another [25].

The fitness score is a measure of how well the solution fits the problem, with higher scores indicating better fit [30].

The differential equation that models the variation in water level in a reservoir depending on the inlet and outlet flows is obtained by following the theoretical derivation steps based on the physical principles of volume conservation and Torricelli’s law for the outlet flow.

According to the law of conservation of volume, the change in the volume of water in the lake over time is the difference between the inlet and outlet flow rates:(1)$$\frac{dV(t)}{dt}=Q_{o}\left(t\right)-Q_{i}\left(t\right)$$

For a tank with constant surface area $A_{b}$,(2)$$V\left(t\right)=A_{b}\cdot H\left(t\right)\to \frac{dV\left(t\right)}{dt}=A_{b}\cdot \frac{dH\left(t\right)}{dt}$$

From relations (1) and (2) it follows that(3)$$Q_{o}\left(t\right)-Q_{i}\left(t\right)=A_{b}\cdot \frac{dH\left(t\right)}{dt}$$

According to Torricelli’s law, the outlet flow rate through an orifice located at the base of a column of liquid is(4)$$Q_{o}\left(t\right)=A_{out}\left(t\right)\cdot \sqrt{2\cdot g\cdot H(t)}$$

In the case of a rectangular opening, the exit area is(5)$$A_{out}\left(t\right)=c\cdot e(t)$$

From relations (4) and (5) it follows that$$Q_{o}\left(t\right)=c\cdot e(t)\cdot \sqrt{2\cdot g\cdot H(t)}$$

The differential Equation (6) describes the variation in the level as a function of the inlet and outlet flow rates. E is derived from the conservation of volume and Torricelli’s law. Substituting Qo(t) in Equation (3) yields(6)$$\frac{dH\left(t\right)}{dt}\cdot A_{b}-c\cdot e\left(t\right)\cdot \sqrt{2\cdot g\cdot H\left(t\right)}+Q_{i}=0$$
where

$\frac{dH(t)}{dt}$ represents the rate of variation in the level at time t;$H\left(t\right)$ is the time-varying level of the lake;$A_{b}$ is the surface of the reservoir;c is the section constant of the beam (its width);$e\left(t\right)=v\cdot t$ is the opening of the sluice;v=0.03 m/s is the operating speed of the sluice;$Q_{i}$ represents the measured input flow.

To determine the transfer function $\frac{H(s)}{e(s)}$ the following steps must be followed:Identifying the equilibrium point;Linearizing the equation;Applying the Laplace transform;Determining the transfer function;Identifying the equilibrium point.

Suppose that$${H\left(t\right)=H}_{0}+h(t)$$$$e\left(t\right)=e_{0}+\delta \cdot e(t)$$$$Q_{i}=Q_{i0}\;(\mathrm{constant})$$

The equilibrium point is given when $\frac{dH}{dt}=0$; that is$$-c\cdot e_{0}\sqrt{2\cdot g\cdot H_{0}}+Q_{i0}=0\to Q_{i0}=c\cdot e_{0}\sqrt{2\cdot g\cdot H_{0}}$$

Linearizing the equation

We write the equation with variations$$A_{b}\frac{d(H_{0}+h)}{dt}-c\cdot \left[e_{0}+\delta \cdot e\left(t\right)\right]\cdot \sqrt{2\cdot g\cdot \left(H_{0}+h\right)}+Q_{i0}=0$$

We approximate the radical with a first-order Taylor expansion:$$\sqrt{H_{0}+h}\approx \sqrt{H_{0}}+\frac{1}{2\sqrt{H_{0}}}\cdot h$$

We replace and keep only the linear terms:$$A_{b}\cdot \frac{dh(t)}{dt}-c\cdot \{e_{0}[\sqrt{2\cdot g\cdot H_{0}}+\frac{g}{\sqrt{2\cdot g\cdot H_{0}}}h\left(t\right)]+\delta \cdot e\left(t\right)\cdot \sqrt{2\cdot g\cdot H_{0}}\}+Q_{i0}=0$$

We know that $Q_{i0}=c\cdot e_{0}\sqrt{2\cdot g\cdot H_{0}}$, so we can simplify:$$A_{b}\cdot \frac{dh(t)}{dt}-[c\cdot e_{0}\cdot \frac{g}{\sqrt{2\cdot g\cdot H_{0}}}h\left(t\right)+c\cdot \sqrt{2\cdot g\cdot H_{0}}\cdot \delta \cdot e\left(t\right)]=0$$

Applying the Laplace transform to null initial conditions results in$$A_{b}\cdot s\cdot H(s)-[\frac{c\cdot e_{0}\cdot g}{\sqrt{2\cdot g\cdot H_{0}}}\cdot H\left(s\right)+c\cdot \sqrt{2\cdot g\cdot H_{0}}\cdot E\left(s\right)]=0$$

Determining the transfer function yields$${(A}_{b}\cdot s+\frac{c\cdot e_{0}\cdot g}{\sqrt{2\cdot g\cdot H_{0}}})\cdot H\left(s\right)=c\cdot \sqrt{2\cdot g\cdot H_{0}}\cdot E\left(s\right)$$

Result in Figure 8 the transfer function is(7)$$\frac{H(s)}{e(s)}=\frac{c\cdot \sqrt{2\cdot g\cdot H_{0}}}{A_{b}\cdot s+\frac{c\cdot e_{o}\cdot g}{\sqrt{2\cdot g\cdot H_{0}}}}$$
where

$A_{b}$ = 10,000 $\left[m^{2}\right]$;c = 1 [m];g = 9.81 $\left[\frac{m}{s^{2}}\right]$;$H_{0}$ represents the initial level of the lake = 70 [m];$e_{0}$ represents the initial opening of the sluice = 0.05 [m].

The program for optimizing the PID controller with Particle Swarm optimization was written in Python.

After executing the program, the following values were obtained:

Kp = 10.1;Ki = 7.56;Kd = 0.14.

The number of iterations was halved from 100 to 50 and the upper bound was changed from 30 to 300.

In Figure 9, the two responses are represented as a PID system and Particle Swarm Optimization on the same graph. When the level increases. PSO stabilizes in 8 min compared to PID, which requires 58 min to stabilize.

In Figure 10, the two responses are represented as a PID system and Particle Swarm Optimization on the same graph when the level decreases. PSO stabilizes in 8 min compared to PID, which requires 58 min to stabilize.

In Figure 9 and Figure 10 the safe PID parameters are as follows: Kp = 5.3; Ki = 2.1; Kd = 0.06. The PID parameters for PSO are as follows: Kp = 10.1; Ki = 7.56; Kd = 0.14.

It was discovered that the higher the number of iterations, the better the performance of the system.

In Figure 9, it is shown how the PSO system reacts when the level must increase, and it takes advantage of the inflow efficiently by closing the sluice. In Figure 10, the PSO system opens the sluice 40% faster to decrease the level than the PID system does. From the simulations results, it can be observed that the optimized PID improves the performance of the level control system in a dam, obtaining a minimum stabilization time without overflow and a near-zero stationary error.

### 5.3. Tuning the PID Controller Using Reinforcement Learning in MATLAB

Reinforcement learning is an ideal technique for decision-making in a dynamic environment, where the agent must learn through repeated interactions with the environment and the reservoir in order to optimize long-term actions [31].

Reinforcement learning is an algorithm wherein decisions are rewarded if they are correct; otherwise, penalties are received [32].

Performance is assessed based on the stability of the water level and the ability to react to unexpected events, such as heavy rains. Water level control in the reservoir was achieved using a deep deterministic policy gradient agent.

The PID controller was optimized using the Tune PI Controller approach under reinforcement learning in MATLAB, using a Twin-Delayed Deep Deterministic agent.

In Figure 11, the difference between the measured and set level represents the error.

The model of the water tank system is shown in Figure 12, where the flow entering the reservoir is integrated and the height of the level in the reservoir is obtained. This height influences the flow of water discharged from the reservoir.

To simulate the controller, the simulation time of 10 s and the controller sampling time of 0.1 s were specified.

The controller parameters are as follows: Kp = 5.3; Ki = 2.1; Kd = 0.06.

The adjusted proportional, integral, and derivative gains are roughly defined.

To define the training model of the reinforcement learning agent, the water tank model was modified, as shown in Figure 13.

An agent reinforcement learning block was inserted;The wave vector is created.

In Figure 13, the flow entering the reservoir is integrated, and the height of the level in the reservoir is obtained. This height influences the flow of water discharged from the reservoir.

The reward function for the reinforcement learning agent is defined as the negative of the Linear Quadratic Gaussian cost, as follows (8):

(8)$$\mathrm{Reward}=-\left(Href-h\left(t\right){)}^{2}+0.01\cdot u^{2}\left(t\right)\right)$$
in which

h is the height of water in the tank;Href is the water reference height.

The reinforcement learning agent maximizes this reward, thereby minimizing the cost of the Linear Quadratic Gaussian function.

To create the actor, a basic neural network was developed using TensorFlow. Three criticals are used because each can learn about the gains of both proportional, integral, and derivative constants individually, as they are very different values in adjusting their parameters and must have their own learning in the system.

Table 6 lists the parameters used for the actor and critic networks of a Twin-Delayed Deep Deterministic agent.

Training stops when the agent can control the water level in the tank by receiving an average cumulative reward greater than −355.

The integral, proportional, and derivative gains of the PID controller were the absolute weights of the actor representation.

After 100 episodes, the following PID controller values were obtained: Kp = 7.3; Ki = 6.1; Kd = 0.06

A simulation of the step response of the system was performed using the gains obtained from the reinforcement learning agent.

Several functions were created to determine optimal parameters:Creation of reinforcement learning water storage environment function: Randomize the reference signal and initial reservoir water height at the beginning of each episode.Linearization function and calculation of stability margins of the SISO water storage system.Simple actor and critic neural network creation functions using TensorFlow or PyTorch.

We have the following linear model described by Equations (9) and (10):(9)$$\frac{dx}{dt}=A\cdot x+B\cdot u$$


y = C∙z(10)


The PID controller parameters were tuned using MATLAB’s PID Tuner application. The optimal values of the PID controller are as follows: Kp = 4.75; Ki = 5.9; Kd = 0.03.

The main results and the response characteristics are presented in Table 7.

In Figure 14, the two responses are represented as the PID System Response and Response for reinforcement learning when the level decreases represented on the same graph.

In Figure 15, the two responses are represented as the PID System Response and Response for reinforcement learning when the level increases represented on the same graph.

In Figure 14 and Figure 15, the safe PID parameters are as follows:

Kp = 5.3;

Ki = 2.1;

Kd = 0.06;

The PID parameters for RL are as follows:

Kp = 7.3;

Ki = 6.1;

Kd = 0.06.

In Figure 14 and Figure 15 the blue response represents the first case. From the graph, it can be seen that it approaches the desired value, but does not reach it; it is more aggressive than the response in the second case, with a large overshoot and a slightly shorter response time.

The response represented in red is the response for Twin-Delayed Deep Deterministic tuning. As shown in Figure 10 and Figure 11, the system is slower and does not exhibit any overshoot, ultimately stabilizing precisely at the reference value Href = 70.1.

The benefit of using a Deep Deterministic Policy Gradient, which incorporates a Twin-Delayed Deep Deterministic variant training policy, is that the system adapts continuously, eliminating the need for the user to engage in redesign for applied control.

After achieving satisfactory training performance on the model, the agent was tuned online in the actual process to adapt to real dynamics. The simulations demonstrated the successful use of machine learning algorithms to adjust the PID controller parameters for optimal control of the reservoir level control system. The developed system ensures the optimal parameters for the PID controller and performs automatic level regulation in a reservoir, eliminating flooding in the area where it is located, considering the environmental conditions.

### 5.4. The Physical System Response

Although the control system taken alone reacts very quickly, the physical process has a certain speed limit that cannot be surpassed. There are two main limitations to the reaction speed of the system: the water flow rate and the command of the sluice opening.

Figure 16 shows how quickly the sluice reacts to the opening command.

The slope of the graph represents the opening speed, which is v = 0.03 m/s.

### 5.5. The Performance Evaluation of the Proposed PSO + RL System

To evaluate the performance of the proposed system, trials over three consecutive days were conducted under normal conditions (125 m^3^/s inflow rate) and one day under high disturbances (187 m^3^/s inflow rate) during a heavy rain period and other four tests in four nonconsecutive days when the inflow was above and below the average. Each test lasted for one hour. The obtained results have been compared by analyzing the main performance indexes.

#### 5.5.1. The Performance Evaluation Under Normal Conditions

In Figure 17 and Figure 18, the two responses are represented on the same graphic: response for reinforcement learning + Particle Swarm Optimization and PID System when the level increases and decreases for the 3rd trial day.

The main difference between the studied systems is the stabilization time which varies considerably from the PID to the proposed system; so, the best performance indicators in this case are IAE (Integral Absolute Error), ISE (Integral Squared Error), and ITAE (Integral Time Absolute Error).

In Table 8, there are the measured parameters for the system with PID.

In Table 9 are the calculated performance indexes for PID.

In Table 10 there are the measured parameters for the PSO + RL system.

Table 11 displays the calculated performance indexes for PSO + RL.

In Figure 19, the comparison between the performance indexes for PID and PSO + RL systems is presented when the systems performed under normal conditions.

#### 5.5.2. The Performance Evaluation Under High Disturbances

In the following, the results of the tests under conditions of a flow rate of 187 m^3^/s during torrential rains in the area are shown.

The water level started from the value of 68 m (intentional discharge for the purpose of carrying out the test).

In Figure 20, the behavior of the PID control system is shown at a high input flow.

The PID parameters are as follows:

Kp = 5.3;

Ki = 2.1;

Kd = 0.06.

The system behavior when the PSO + RL control system was used is shown in Figure 21.

The PID parameters found by using PSO + RL are as follows:

Kp = 10;

Ki = 0.08;

Kd = 0.

To highlight the performance improvement through PSO + RL, Figure 22 shows the overlapped graphs for both versions.

The evaluation parameters for the two control systems are shown below.

Table 12 displays the calculated performance indexes for PID and PSO + RL.

In Figure 23, the performance indexes for PID and PSO + RL systems are shown.

According to the results presented above, the combination of Particle Swarm Optimization and reinforcement learning offers a considerably better control than the stand-alone PID.

### 5.6. Robustness and Reproducibility Analysis

#### 5.6.1. Robustness Evaluation

To demonstrate the robustness of the proposed PSO + RL control system, a set of sensitivity analyses and stress tests were performed:Sensitivity to inflow rate variations

The inflow rate Qi was varied by ±10% and ±20% around the nominal value of 125 m^3^/s. The results showed that the system remained stable without exceeding the safety limit of 72 m, while maintaining settling times below 2 min in all cases.

Table 13 displays the behavior of the PSO + RL system when the inflow varies as stated above.

Robustness to measurement noise

Gaussian noise with zero mean and standard deviation σ = 0.07 m was added to the water level measurements. Despite the disturbances, the PSO + RL system maintained stable operation and a settling time below 2 min.

Response under extreme disturbance

During high inflow events (187 m^3^/s), the PSO + RL system successfully avoided overflow and reached steady state in less than 1.8 min, compared with more than 6 min for the classical PID as shown in Figure 21.

#### 5.6.2. Reproducibility Evaluation

To ensure the reproducibility of the proposed methodology, several measures were taken:Fixed random seed: All PSO and RL simulations were executed using a fixed random seed (np.random.seed(42)), ensuring identical initial conditions across independent runs.Multiple runs: The optimization was repeated in 50 independent runs. The mean and standard deviation of the main performance indexes were calculated.Full parameter disclosure: All PSO hyperparameters (number of particles = 20; inertia weight = 0.7; cognitive coefficient = 1.5; social coefficient = 1.5) and RL training parameters (actor learning rate = 0.01; critic learning rate = 0.01; batch size = 128; training episodes = 100) are reported.Code availability: The complete Python source code for PSO is provided in the Appendix A.

## 6. Computational Issues

The hybrid PSO + RL strategy developed is computationally demanding, particularly for online control applications. However, no indication of computational requirements is provided in the paper nor are hardware limitations, inference time, or fallback techniques for online applications.

### 6.1. Computational Requirements

The proposed hybrid PSO + RL control strategy introduces additional computational requirements compared with classical PID controllers. Since the system is intended for online control applications in critical hydraulic infrastructures, an analysis of computational feasibility was performed

#### 6.1.1. Hardware Platform

All simulations and training were carried out on the following hardware configuration:CPU: Intel Core i7-11700K @ 3.6 GHz (8 cores, 16 threads);GPU: NVIDIA RTX 3070 with 8 GB VRAM, CUDA 11.4 support;RAM: 32 GB DDR4;Software environment: Python 3.10 with TensorFlow 2.12 and MATLAB R2023a.

##### Training and Optimization Performance Are Summarized as Follows

PSO Time: The PSO module required approximately 8 min for 50 iterations to provide initial PID parameters guiding the RL agent;RL Training Time: The RL agent required approximately 2 h for 100 episodes to achieve stable performance with cumulative reward > −355;Inference Latency: After training, the control policy delivered actions in 4.2 ms per step, well below the control loop sampling time of 0.1 s, ensuring real-time feasibility.

##### Resource Utilization

During online operation the following was observed:CPU usage averaged 12%;GPU usage remained below 5%;Memory consumption was less than 4 GB RAM, leaving sufficient resources for additional monitoring tasks.

#### 6.1.2. Fallback and Safety Mechanisms

To address the risk of computational delays or inference anomalies in online control, the following measures were implemented:A safe PID controller (Kp = 5.3; Ki = 2.1; Kd = 0.06) operates as a fallback mechanism, automatically taking control if the RL inference exceeds 50 ms or produces invalid outputs;The sluice opening speed is physically limited to v = 0.03 m/s, preventing dangerous commands regardless of the controller decision;A watchdog system continuously monitors inference latency and performance indexes to trigger fallback when necessary.

In Table 14, the summary of the installed computational characteristics, the training and optimization time for RL and PSO, the average response time for RL and its sampling period, the percentage of charge for the computer CPU and GPU when performing the task, and the fallback strategy are listed.

### 6.2. Conclusions

The computational evaluation demonstrates that the hybrid PSO + RL control strategy is feasible for online deployment. With inference times well below the sampling interval and a robust fallback mechanism, the method can be safely applied in real-world dam operation scenarios while ensuring high performance and resilience.

## 7. Practical Feasibility and Limitations

Although the integration of the hybrid PSO + RL method has shown promising results in simulations and initial implementation in the dam, it is essential to recognize and discuss the practical limitations and challenges for large-scale application:Training data requirements and quality:

For RL to learn effective policies, extensive data sets, representative of variable hydrological conditions, are required. The quality and accuracy of sensor data are critical, and noise or measurement errors can degrade the performance of the algorithm.

Robustness to noise and unexpected disturbances:

In the real environment, hydraulic systems are subject to sudden disturbances and unexpected fluctuations. The algorithm must be robust enough not to generate unstable or overly aggressive commands that can damage equipment or cause dangerous situations.

Computational constraints and latency:

The online tuning of the RL controller involves significant computational requirements. In the context of critical systems, processing latency must be minimal to ensure real-time responses, and hardware resources available in the industrial environment must be sufficient.

Risks associated with online tuning:

Continuous adaptation of controller parameters in real time, if not managed correctly, can introduce instability, especially in systems where safety is critical. A robust fallback mechanism to classical methods in case of anomaly or degraded performance was put in place.

Need for extensive validation:

For a production implementation, extensive long-term validation, including tests under varied climatic and hydrological conditions, is imperative to ensure system stability and reliability.

## 8. Conclusions

The integration of Particle Swarm Optimization (PSO) with reinforcement learning (RL) has proven to be an effective hybrid strategy, combining the exploratory search and global optimization capabilities of PSO with the adaptive, model-free decision-making power of RL. While PSO efficiently identified a near-optimal set of parameters for initializing the RL agent, RL continuously refined the control policy based on real-time interactions with the system environment. This synergy led to improved responsiveness, reduced flooding risk, and optimized reservoir usage, as demonstrated through both simulations and real-world deployment.

In conclusion, this study confirms that the hybrid RL + PSO approach not only increases system intelligence and control precision but also lays the foundation for autonomous, sustainable, and resilient water resource management systems, crucial in the context of accelerating climate variability.

## 9. Future Perspectives

It was found that to achieve the desired performance in dynamic and uncertain hydrological environments, it is preferable not to rely on a single control method. Instead, combining intelligent control systems can better address the specific requirements of each real-world application.

In particular, reinforcement learning (RL) assisted by Particle Swarm Optimization (PSO), and guided by a simplified mathematical model of the system, has proven to be a viable and efficient solution. PSO provides optimized initial policy parameters or hyperparameters, which significantly reduce the training time and improve convergence of the RL agent. This hybrid strategy becomes particularly advantageous in nonlinear and partially observable systems, such as water reservoirs under climate uncertainty.

Building on the promising results obtained, future research and development directions include the following:Integration of weather forecasting into the RL agent: Incorporating short- and medium-term meteorological predictions (e.g., precipitation forecasts, temperature, inflow estimations) will enable more proactive and anticipatory decision-making, especially in the face of torrential rains or prolonged droughts.Scalability and deployment across multiple reservoirs: The current system can be scaled and adapted for real-time coordination between multiple dams in Romania, improving water distribution, energy efficiency, and flood protection at a regional level.Transition to online reinforcement learning: Future systems could include online learning agents that continue to improve based on new environmental data, ensuring adaptability to climate change and anthropic interventions.Enhancing robustness and cybersecurity: With increased autonomy and remote operation, the system should integrate fault-tolerant mechanisms and cybersecurity protocols to maintain safe and reliable operation of critical infrastructure.Extension to multi-objective control: Further developments could target not only flood prevention and level regulation, but also hydropower optimization, environmental flow preservation, and sediment control in dam reservoirs.

In conclusion, the intelligent automation system developed in this project has the potential to be generalized and implemented on a national scale, offering a flexible, adaptive, and sustainable approach for modern dam management in Romania and beyond.

## Figures and Tables

**Figure 1 sensors-25-05055-f001:**
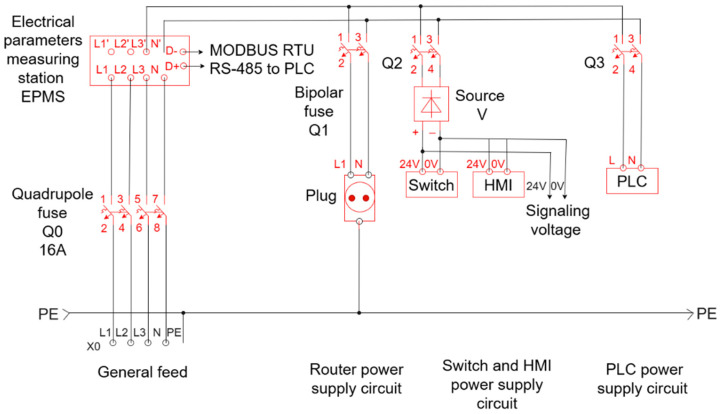
Electrical diagram.

**Figure 2 sensors-25-05055-f002:**
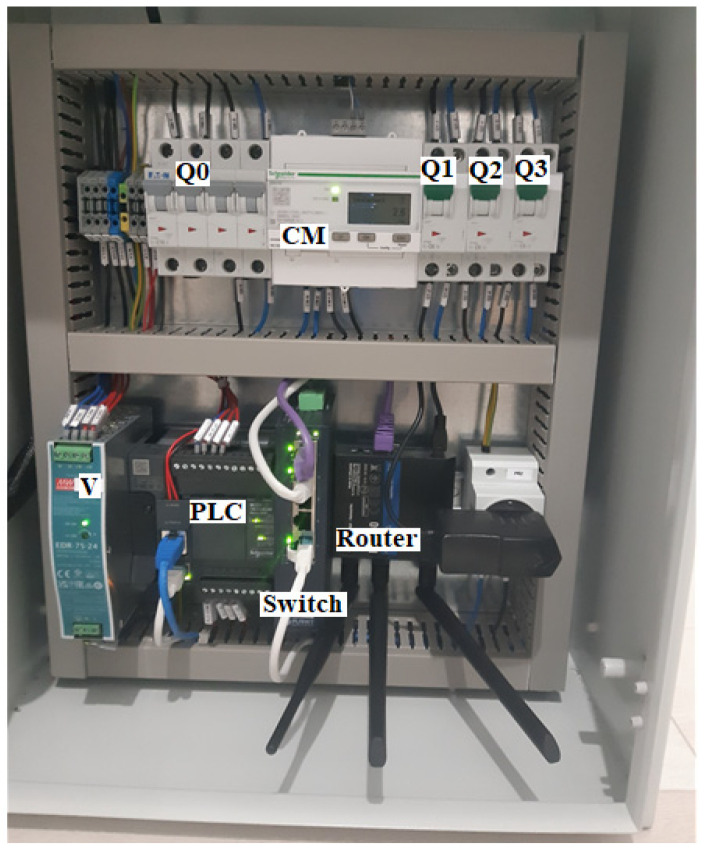
Components of the automation panel.

**Figure 3 sensors-25-05055-f003:**
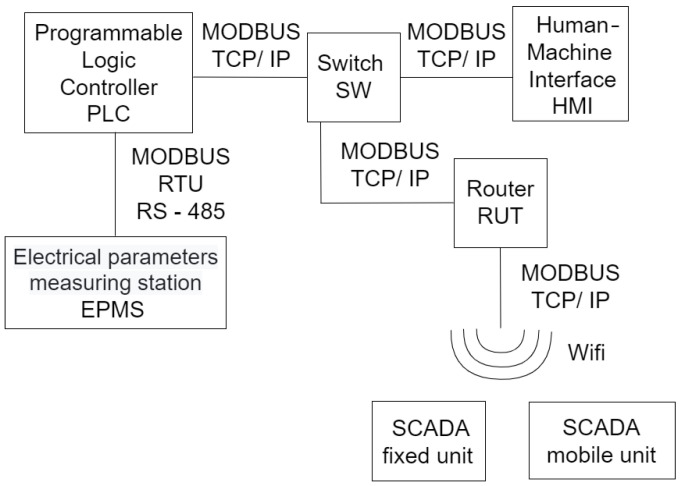
SCADA architecture.

**Figure 4 sensors-25-05055-f004:**
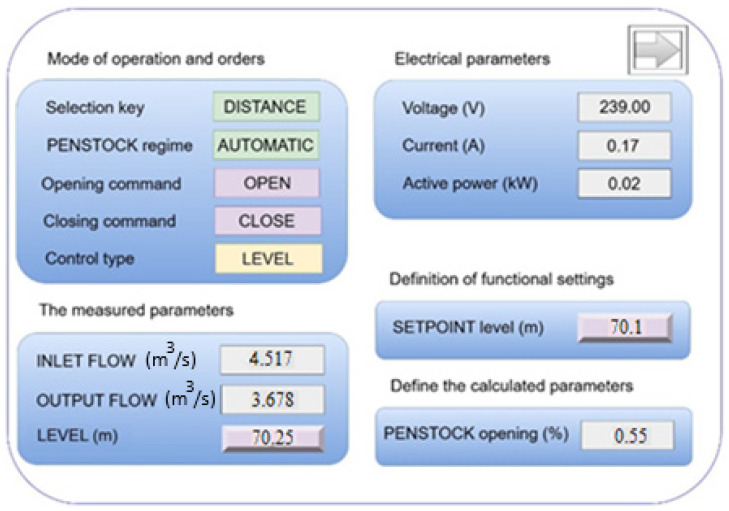
HMI for level adjustment.

**Figure 5 sensors-25-05055-f005:**
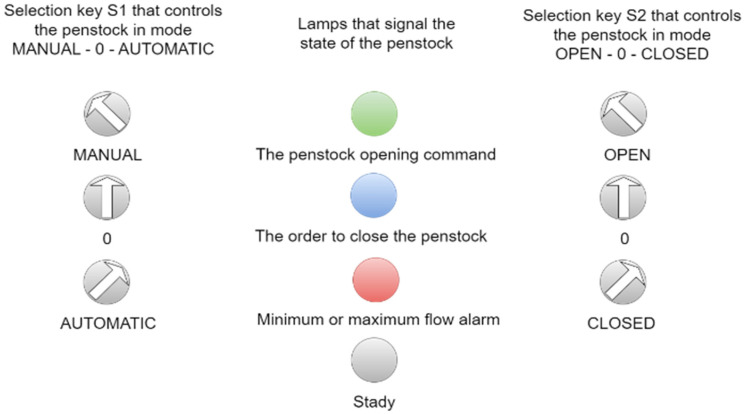
Selection keys that control the controls and status of the sluice displayed on the electrical panel.

**Figure 6 sensors-25-05055-f006:**
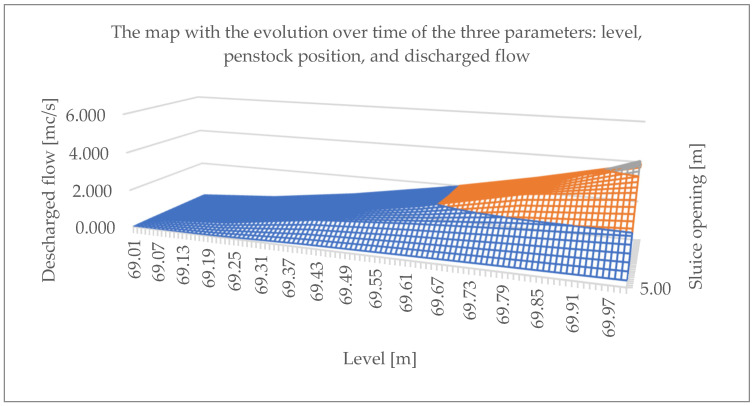
The map with the evolution over time of the three parameters: level, sluice opening, and discharged flow.

**Figure 7 sensors-25-05055-f007:**
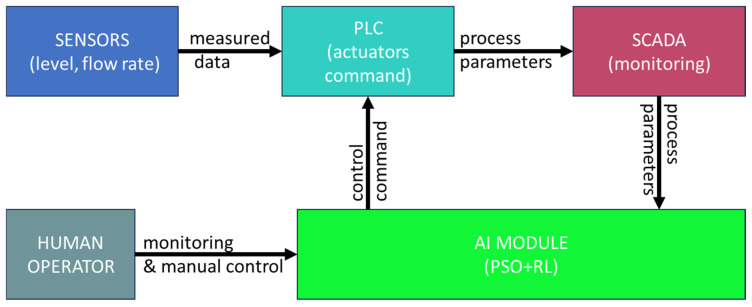
The control system architecture.

**Figure 8 sensors-25-05055-f008:**
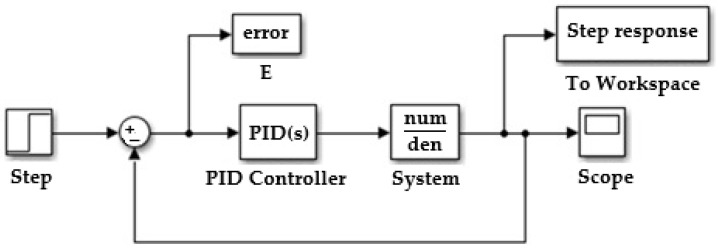
Basic diagram of the PID control system.

**Figure 9 sensors-25-05055-f009:**
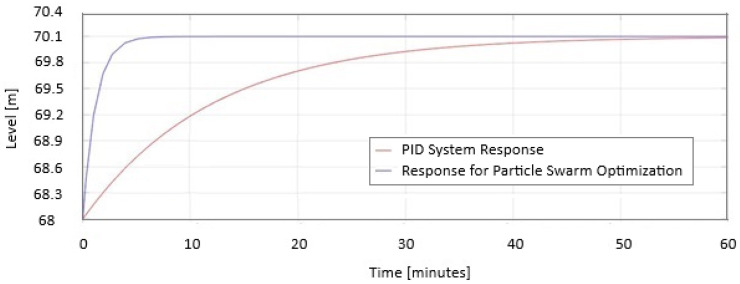
The PID and Particle Swarm Optimization responses are displayed on the same graph when the level increases.

**Figure 10 sensors-25-05055-f010:**
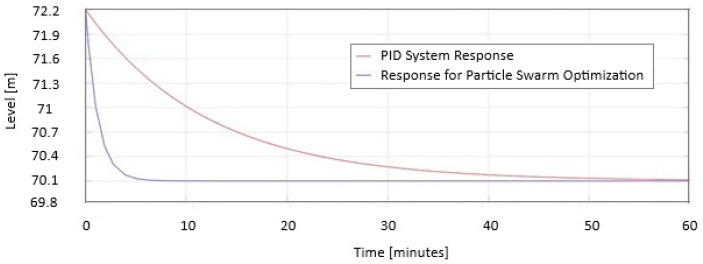
The PID and Particle Swarm Optimization are displayed on the same graph when the level decreases.

**Figure 11 sensors-25-05055-f011:**
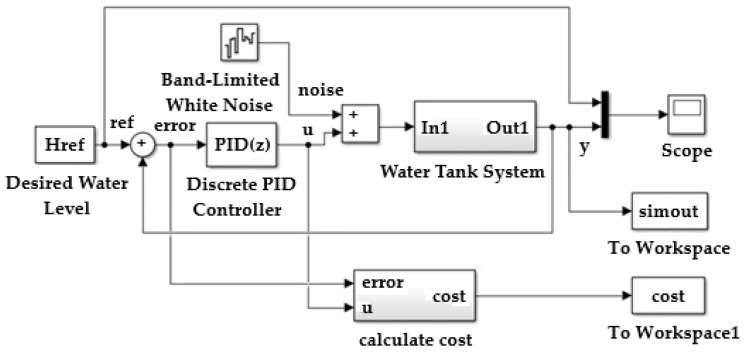
Controlling the water level in a reservoir using a deep deterministic policy.

**Figure 12 sensors-25-05055-f012:**
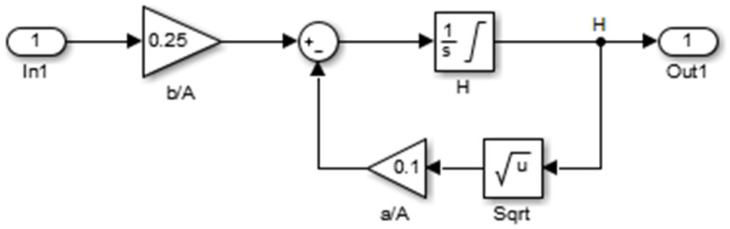
Model from the water tank system.

**Figure 13 sensors-25-05055-f013:**
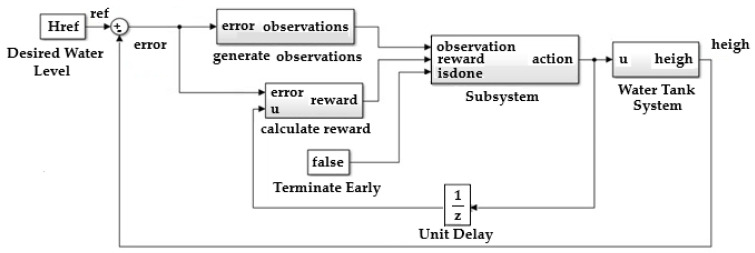
MATLAB/Simulink implementation block diagram of the proposed control system using reinforcement learning Twin-Delayed Deep Deterministic agent.

**Figure 14 sensors-25-05055-f014:**
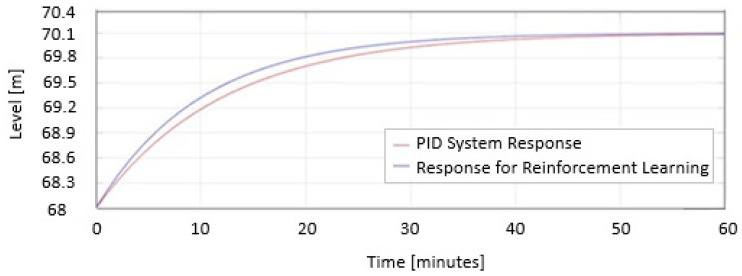
System response in MATLAB/Simulink using reinforcement learning Twin-Delayed Deep Deterministic Agent when the level decreases.

**Figure 15 sensors-25-05055-f015:**
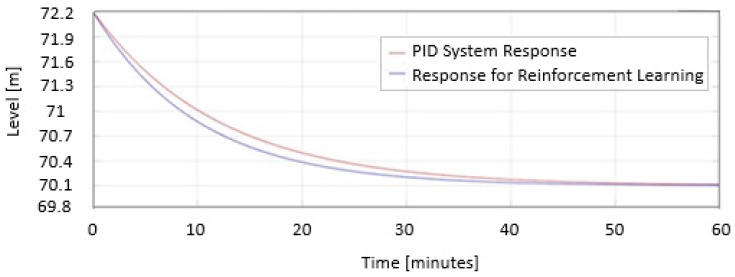
System response in MATLAB/Simulink using reinforcement learning Twin-Delayed Deep Deterministic Agent when the level increases.

**Figure 16 sensors-25-05055-f016:**
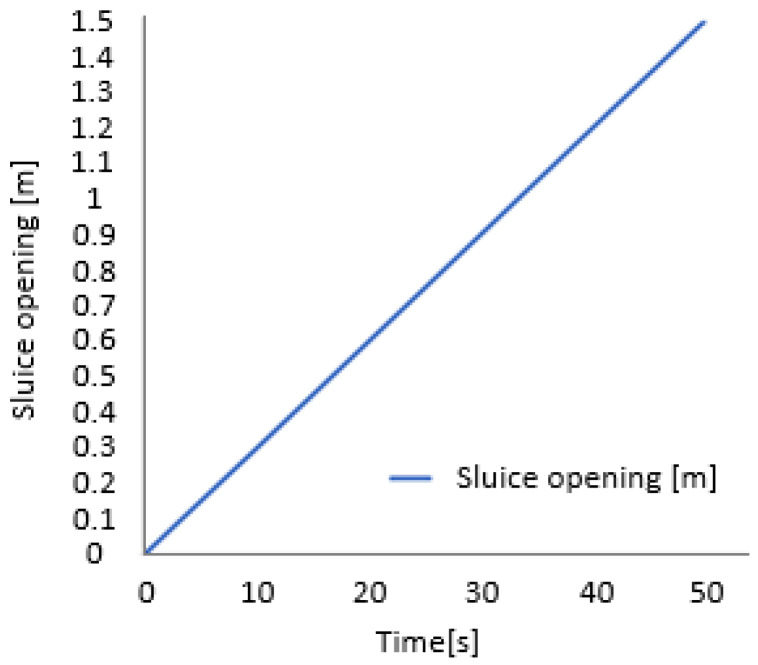
Sluice opening time from closed to fully open.

**Figure 17 sensors-25-05055-f017:**
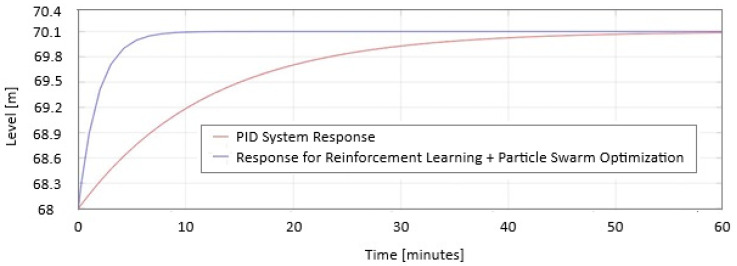
Response for reinforcement learning + Particle Swarm Optimization and PID System Response when the level increases.

**Figure 18 sensors-25-05055-f018:**
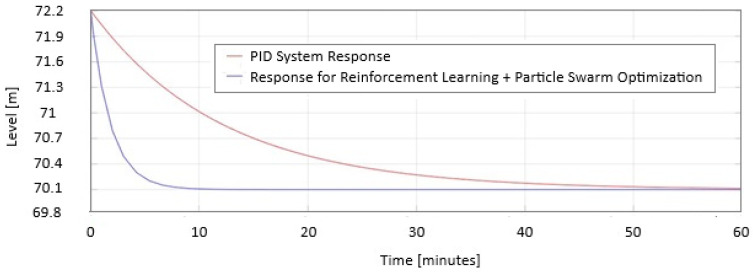
Response for reinforcement learning + Particle Swarm Optimization and PID System Response when the level decreases.

**Figure 19 sensors-25-05055-f019:**
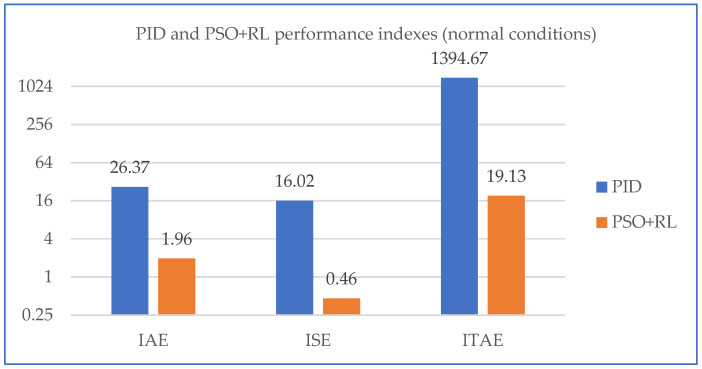
Response PID and PSO + RL performance indexes under normal conditions (125 m^3^/s).

**Figure 20 sensors-25-05055-f020:**
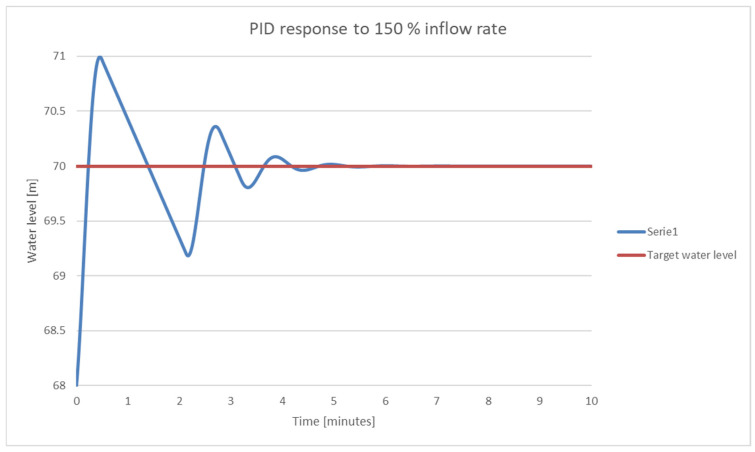
PID response for 187 m^3^/s input flow.

**Figure 21 sensors-25-05055-f021:**
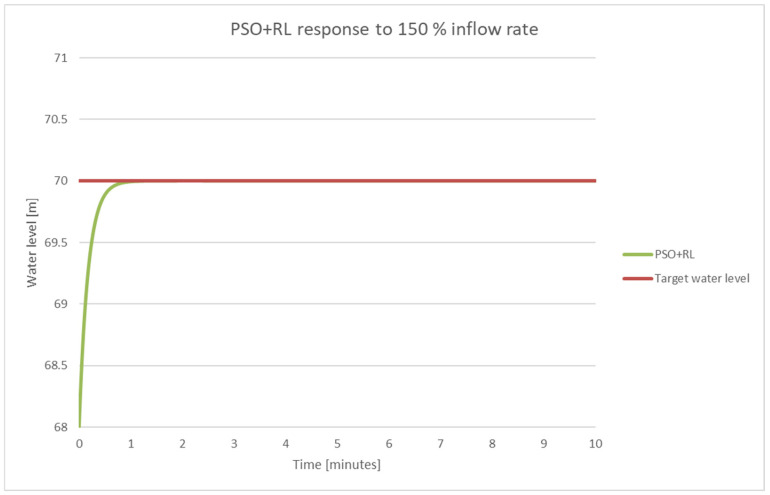
PSO + RL response for 187 m^3^/s input flow.

**Figure 22 sensors-25-05055-f022:**
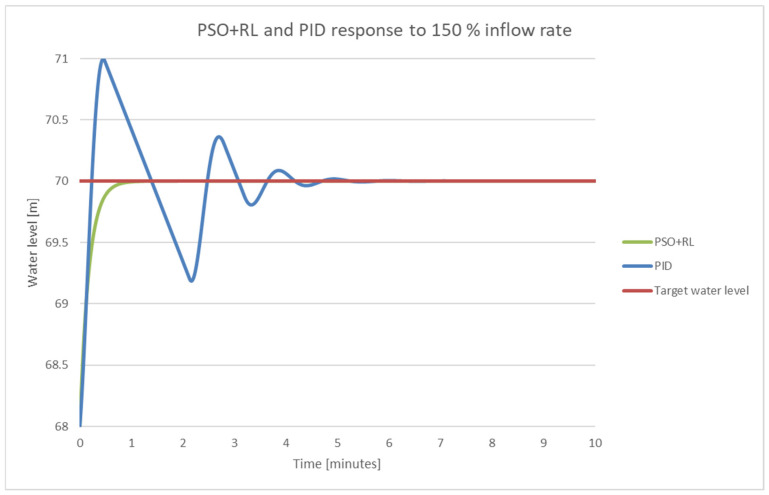
PSO + RL and PID response for 187 m^3^/s input flow.

**Figure 23 sensors-25-05055-f023:**
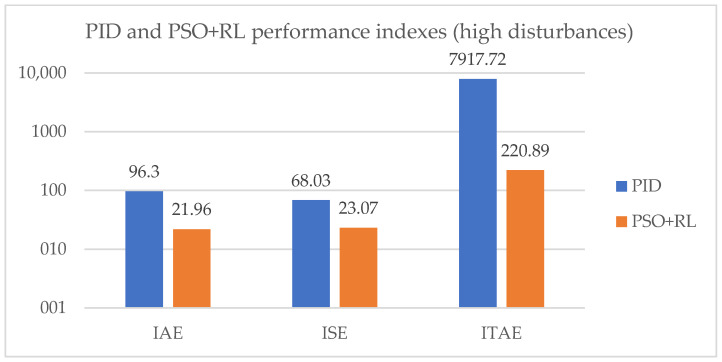
PSO + RL and PID performance indexes for 187 m^3^/s input flow.

**Table 1 sensors-25-05055-t001:** Comparison between Output Feedback Adaptive and PSO + RL methods.

Criterion	Output Feedback Adaptive Optimal Control	Particle Swarm Optimization + Reinforcement Learning
Type method	Feedback-based control + adaptation	Global optimization + learning
Real-time control?	Yes, for initial design	PSO is more suitable for offline optimization
Dynamic adaptation?	Yes—adaptive online control	No—PSO optimizes static parameters or offline policies
Complexity	Yes—go on	Large (optimization + learning)
Theoretical stability	Medium to large (nonlinear equations + adaptation)	No strict theoretical guarantee
Sturdiness	It can be proved formally (Lyapunov)	Excellent in global exploration

**Table 2 sensors-25-05055-t002:** Comparative advantages of the two methods.

Appearance	Model-Free Output Feedback Control	PSO + RL
Model independence	Completely model-independent	PSO may require an approximate model for guidance
Convergence speed	Fast if the data is good	PSO accelerates RL → faster convergence than simple RL
Response quality	Good for tracking a reference in a 2D system	Very good when PSO is well guided by a mathematical model
Parameter optimization	Does not optimize RL hyperparameters	Yes—optimize policies, networks, or other RL parameters
Sturdiness	Good against process variations (through data learning)	Excellent—looks for good solutions globally, avoids getting stuck in local minima
Appearance	Model-free Output Feedback control	PSO + RL

**Table 3 sensors-25-05055-t003:** Comparative disadvantages of the two methods.

Appearance	Model-Free 2D Tracking	PSO + RL
Need data	Strongly dependent on historical data	Less dependent— PSO can explore and learn
Computational complexity	Medium to large	Large—PSO and RL combined involves multiple evaluations per iteration
Guaranteed stability?	Not under all conditions	Not exactly—PSO optimizes performance, does not ensure formal stability
Exploration vs. exploitation	Based on RL policies	PSO can get stuck in local optimum without diversification
Sensitive tuning	Requires correct choice of reward function	Wrong choice of PSO parameters affects performance

**Table 4 sensors-25-05055-t004:** The values of the PID controller parameters.

Controller Type	Kp	Ti	Td
PID	4	5	0.1

**Table 5 sensors-25-05055-t005:** Comparison of the optimization techniques used for adjusting the PID controller parameters: reinforcement learning versus Particle Swarm Optimization [25].

Characteristics	Reinforcement Learning	Particle Swarm Optimization
Optimization method	Trial and error, based on rewards	Search by swarm behavior
Application	Sequential and complex problems	Continuous and uncomplicated problems
Efficiency	Effective in continuous learning	Fast for simple functions
The risk of blocking	Exploration-exploitation	Local minima
Convergence	Requires a lot of data; converges slowly	Fast convergence
Complexity	Complex and high computational resources	Easy to implement

**Table 6 sensors-25-05055-t006:** Training parameters for actor and critic networks.

The Parameters	Value
Mini size—lot	128 samples
Gaussian noise variation	0.1
The fully connected learning dimension	32
Actor learning rate	0.01

**Table 7 sensors-25-05055-t007:** Adjusted parameters and system performance.

Tuned Parameters and Step Response Characteristics	Kp	Ki	Kd	Overpass [%]	Time Settling [h]
Tuning based on TD3	7.3	6.1	0.06	0	0.3
Adjustment based on linearizers	106	5.9	0.03	1	0.1

**Table 8 sensors-25-05055-t008:** Measured parameters for the PID system.

Day	Target Level [m]	Obtained Level [m]	Stabilization Time [min]	Level Oscillations [m]
1	70	69.2	50	±0.8
2	70	69.5	55	±0.5
3	70	69.8	58	±0.2

**Table 9 sensors-25-05055-t009:** Calculated performance indexes for PID.

Day	Stationary Error [m]	Stabilization Time [min]	IAE	ISE	ITAE
1	0.8	50	40	32	2000
2	0.5	55	27.5	13.75	1512
3	0.2	58	11.6	2.32	672

**Table 10 sensors-25-05055-t010:** Measured parameters for the PSO + RL system.

Day	Target Level [m]	Obtained Level [m]	Stabilization Time [min]	Level Oscillations [m]
1	70	69.8	9.8	±0.2
2	70	69.9	10	±0.1
3	70	69.7	9.7	±0.3

**Table 11 sensors-25-05055-t011:** Calculated performance indexes for PSO + RL.

Day	Stationary Error [m]	Stabilization Time [min]	IAE	ISE	ITAE
1	0.2	9.8	1.96	0.392	19.2
2	0.1	10	1	0.1	10
3	0.3	9.7	2.91	0.873	28.2

**Table 12 sensors-25-05055-t012:** The evaluation parameters for the PID and PSO + RL systems.

Control System	IAE [ms]	ISE [m^2^s]	ITAE [ms^2^]
PID	96.3	68.03	7917.72
PSO + RL	21.96	23.07	220.88

**Table 13 sensors-25-05055-t013:** Sensitivity of PSO + RL to inflow rate variations.

Scenario	Inflow Rate [m^3^/s]	Setling Time [min]	ISE	IAE	ITAE	Max Level [m]
Nominal	125	1	0.15	3	3	69.95
+10% Increase	137.5	1.1	0.2376	3.96	4.356	69.94
−10% Decrease	112.5	1.2	0.18	3.6	4.32	69.95
+20% Increase	150	1.3	0.3822	5.46	7.098	69.93
−20% Decrease	100	1.3	0.2808	4.68	6.084	69.94

**Table 14 sensors-25-05055-t014:** Summary of computational requirements.

Metric	Value
Hardware used	Intel i7-11700K, 32 GB RAM, RTX 3070
PSO time	~8 min (50 iterations)
RL training time	~2 h (100 episodes)
Average inference time	4.2 ms per decision
Sampling period	0.1 s
CPU usage (online) ~12% GPU usage (online) ~5% Fallback strategy Safe PID controller	

## Data Availability

The original contributions presented in the study are included in the article; further inquiries can be directed to the corresponding author.

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
