# Peer review of "Combined Particle Swarm Optimization and Reinforcement Learning for Water Level Control in a Reservoir"

_sensors, 2025, doi:10.3390/s25165055_

Round 1
Reviewer 1 Report
Comments and Suggestions for Authors
1. Could the authors provide more details on the limitations of the proposed method? While the improvements in control accuracy are clear, understanding its constraints could help in identifying future research directions.
2. The research gap that the authors are contributing to filling in this paper is not clearly defined. It is suggested to break the introduction section into subsections including related literature, and contributions to make it clearer and improve readability for the readers.
3. In terms of application, it is beneficial for general readers to get familiar with the practical and real-world applications of this work. Please briefly describe some real-world applications in the introduction.
4. The contribution is not clear. What is the difference compared with the existing literatures? What is the main difficulty of the proposed method? For clarity of the contribution summarized at the end of the Introduction, the references should be given for comparison.
5. The programs should be included in the appendix.
6. Some reinforcement Learning methods studied in this paper should be analyzed and compared with other similar studies, which is helpful. For example: “Model-free output feedback optimal tracking control for two-dimensional batch processes”; “Output feedback based adaptive optimal output regulation for continuous-time strict-feedback nonlinear systems”; “Two-dimensional model-free Q-learning-based output feedback fault-tolerant control for batch processes”.
7. Only a comparison with the PID method has been conducted; a comparative study with other recent existing methods should also be carried out.
8. In the simulation comparison, model and controller parameters should be included and explain how to get these parameters of model and controller.
At the end, the reviewer is keen to read your paper again after you address the above comments.
Author Response
Review 1
- Could the authors provide more details on the limitations of the proposed method? While the improvements in control accuracy are clear, understanding its constraints could help in identifying future research directions.
- I have added chapter 6 to the article which includes the limitations
of the proposed solutions.
- We also modified chapter 8 which contains future research directions but not all of them refer to the mentioned limitations because some of the limitations are inevitable due to the chosen control solution and they already heve been adressed.
- The research gap that the authors are contributing to filling in this paper is not clearly defined. It is suggested to break the introduction section into subsections including related literature, and contributions to make it clearer and improve readability for the readers.
- We devided the chapter 1 into seven sections to be easily understandble.
The researche gap we are filling is presented in chapter 1.5 and the original contribution in chapter 1.7
- In terms of application, it is beneficial for general readers to get familiar with the practical and real-world applications of this work. Please briefly describe some real-world applications in the introduction.
- In chapter 1.8. we briefly described some real-world applications in the introduction.
- The contribution is not clear. What is the difference compared with the existing literatures? What is the main difficulty of the proposed method? For clarity of the contribution summarized at the end of the Introduction, the references should be given for comparison.
- In chapter 1.7 we included the original contribution from this study
- The programs should be included in the appendix.
- The programs were not shared in the paper. We belive that the real added value is the idea behind them and not the code itself.
- Some reinforcement Learning methods studied in this paper should be analyzed and compared with other similar studies, which is helpful. For example: “Model-free output feedback optimal tracking control for two-dimensional batch processes”; “Output feedback based adaptive optimal output regulation for continuous-time strict-feedback nonlinear systems”; “Two-dimensional model-free Q-learning-based output feedback fault-tolerant control for batch processes”.
- The answer is in chapter 1.4.
- Only a comparison with the PID method has been conducted; a comparative study with other recent existing methods should also be carried out.
- It is identical to what I answered in point 6.
- In the simulation comparison, model and controller parameters should be included and explain how to get these parameters of model and controller.
- We obtained the RL parameters in chapter 5.3. and the PSO parameters in chapter 5.2

Reviewer 2 Report
Comments and Suggestions for Authors
To address the issue of automatic water level regulation in reservoirs to eliminate flooding in the surrounding area, this paper develops a novel optimization method combining Particle Swarm Optimization (PSO) and Reinforcement Learning (RL) to adjust the parameters of a PID controller. The effectiveness of the proposed method is validated through simulations using MATLAB and Python, as well as practical applications in the Mariselu Reservoir, Romania. The authors need to make the following revisions:
1.The paper mentions that a mathematical equation guides PSO but does not clarify the specific derivation process of this equation. It is necessary to supplement the detailed formula derivation or cite relevant theoretical support.
2.The paper states that RL requires large-scale data training but does not discuss the data acquisition methods. It is essential to add explanations related to data collection and preprocessing.
3.While the paper mentions successful flood prevention, it does not quantify the flood control effects. The authors should further elaborate with specific metrics.
4.The conclusion states that the system integrates the advantages of the two methods but lacks elaboration. Additionally, the paper lacks an introduction to future research directions, which should be supplemented.
Author Response
Review 2
- The paper mentions that a mathematical equation guides PSO but does not clarify the specific derivation process of this equation. It is necessary to supplement the detailed formula derivation or cite relevant theoretical support.
- Please see chapter 5.2
- The paper states that RL requires large-scale data training but does not discuss the data acquisition methods. It is essential to add explanations related to data collection and preprocessing.
RL was not used in the classic way in which the Agent is trained by the reward functions when exploring the database provided.
In my application, the RL Agent is trained by PSO to converge faster to the solution.
The data acquisition methods used in the work are:
- SCADA: automatic collection of real-time operational data from sensors for: level, flow and valve openings.
- Numerical simulation: if there is not enough historical dam operation data (levels, flows, gate openings), we built a simple mathematical model based on a differential equation that provides the starting point for PSO and RL.
For more details please refer to chapter 5.3
- While the paper mentions successful flood prevention, it does not quantify the flood control effects. The authors should further elaborate with specific metrics.
- Please see chapter 5.5.2
- The conclusion states that the system integrates the advantages of the two methods but lacks elaboration. Additionally, the paper lacks an introduction to future research directions, which should be supplemented.
- For combined advantages elaboration please see chapter 5
-For future research directions please see chapters 7 and 8

Reviewer 3 Report
Comments and Suggestions for Authors
This manuscript presents an integrated control strategy combining Reinforcement Learning (RL) and Particle Swarm Optimization (PSO) to tune a PID controller for water level regulation in a reservoir. The proposed approach is motivated by the limitations of traditional PID tuning methods under dynamic environmental conditions. The authors simulate and implement the strategy using MATLAB, Python, and SCADA systems in a real dam (Mărișelu Dam, Romania), and claim improvements in responsiveness and adaptability for flood prevention. The topic is relevant to both control engineering and water resource management, and the integration of AI methods into a real-world system adds practical value. However, several aspects of the manuscript require clarification, restructuring, or deeper analysis to improve its scientific rigor and presentation quality.
- Clarity of Novelty: The manuscript lacks a clear and concise statement of what is technically novel about the proposed PSO + RL method compared to prior work. Please clarify how this approach differs from hybrid methods already published in the literature (e.g., PSO-based RL policy search, policy gradient methods enhanced by evolutionary techniques, etc.).
- Reference Quality and Depth: The literature review includes some relevant papers but is mostly descriptive. It does not critically assess existing hybrid RL or PSO methods in control or dam systems. Please improve this section to highlight research gaps that justify your work.
- Method Integration: The manuscript refers to a “mathematical equation guiding PSO,” but this guidance function is not clearly defined or contrasted with classical PSO. What exactly is the role of the “mathematical formula” in reducing the search space or improving convergence?
- Limited Discussion of Related Control Strategies: The manuscript overlooks several existing control approaches specifically developed for overflow prevention in storage tanks and reservoirs. For example, some methods integrate energy price signals or multi-objective optimization (e.g., “Energy price as an input to fuzzy wastewater level control in pump storage operation”). I recommend that the authors broaden their literature review to include alternative control philosophies beyond PID tuning, especially those that incorporate environmental, economic, or policy constraints into the control design.
- Missing Unit for Water Level in Figures: Several figures (e.g., Figures 4, 6, 10, 12, and 13) plot the water level variable, but the unit of measurement is not clearly indicated. While some tables suggest the unit might be meters (e.g., “LEVEL [m]”), the figures themselves either omit this information or display a scale from 0 to ~70 without proper labeling. Please ensure all axes in all figures clearly indicate the unit (e.g., “Level [m]”) to avoid confusion.
- Simulation and Real Implementation: While the manuscript claims the method was tested and implemented, there is little distinction between simulated and physical results. Were the PSO+RL-tuned parameters used directly in the real system? How was performance evaluated in the real dam?
- Unrealistic Dynamics in Simulation Results: Some simulation results (e.g., Figures 10, 12, and 13) show the reservoir water level varying from approximately 0 to 70 (presumably meters) within just a few seconds. Such rapid variations are physically unrealistic for real reservoirs or dams, regardless of size. This suggests that the simulation model does not accurately reflect the inertia and hydraulic dynamics of actual water systems. The authors should either justify these time scales (e.g., by clarifying units and scaling factors) or revise the model to better represent real-world physical constraints.
- Lack of Information on the Manipulated Variable: The manuscript does not show or analyze the manipulated variable (e.g., gate position, flow rate adjustment, or control signal). This is a critical omission, as the feasibility of the control strategy depends not only on the system response but also on how rapidly and frequently the actuators must operate. Without this information, it is impossible to assess whether the required control actions are realistic or physically achievable in a dam setting. Please include a plot and analysis of the manipulated variable to demonstrate that the proposed control law results in implementable actuator behavior.
- Unrealistic Test Scenario — Lack of Disturbance Rejection Evaluation: The control performance evaluation is based solely on setpoint tracking tests, where the water level is adjusted to match predefined values. However, in real-world dam operations, the primary challenge is disturbance rejection, such as sudden changes in inflow due to rainfall or upstream releases. The authors should include tests with realistic inflow disturbances to evaluate how well the proposed control system handles unexpected hydraulic events — which is crucial for flood prevention and operational safety.
- Controller Comparison: The comparison between PID, PSO-PID, RL-PID, and the hybrid method is helpful, but it is limited to step response metrics. Please consider including additional performance metrics (e.g., IAE, ISE, robustness to disturbance).
- Code Readability: Several code blocks (particularly Python snippets) contain formatting issues, undefined variables (e.g., simulate_system vs. simulatesystem), or missing context. Please ensure that the code is syntactically correct, complete, and reproducible.
- Figure Captions and Quality: Figures such as 9, 10, 12, and 13 are referred to but are poorly integrated. Captions should be expanded to clearly explain what is being shown, and visual clarity should be improved for readability (e.g., axis labels, units, legends).
- Language and Grammar: The manuscript contains several grammatical issues and awkward sentence structures (e.g., “Phyton” instead of “Python,” “regulator” instead of “controller” in some places). A thorough language revision is necessary.
- SCADA Integration: The SCADA implementation is described, but it is not clear how the real-time controller tuning interacts with the SCADA system. Is the AI module running inside the PLC, remotely, or just for offline tuning?
- Practical Feasibility and Limitations: The conclusion would benefit from a critical discussion of limitations, such as training data requirements, robustness to sensor noise, computational constraints, and potential risks of online tuning in safety-critical systems like dams.
- Citation Style and Editorial Formatting: Several references appear with inconsistent styles or missing information. Please ensure all references are complete and follow MDPI citation standards.
- Inconsistent Use of Decimal Separator: In Figure 4, both the comma and the point are used as decimal separators (e.g., "70,25" and "70.1"), which is confusing and not compliant with English-language conventions. Please revise all numeric values in the manuscript and figures to consistently use the point as the decimal separator, as required in English scientific writing.
Author Response
Review 3
- Clarity of Novelty: The manuscript lacks a clear and concise statement of what is technically novel about the proposed PSO + RL method compared to prior work. Please clarify how this approach differs from hybrid methods already published in the literature (e.g., PSO-based RL policy search, policy gradient methods enhanced by evolutionary techniques, etc.).
- I added at the end of subchapter 1.7.4
- Reference Quality and Depth: The literature review includes some relevant papers but is mostly descriptive. It does not critically assess existing hybrid RL or PSO methods in control or dam systems. Please improve this section to highlight research gaps that justify your work.
-The answer can be found in Chapter 1, subchapter 1.3. and 1.5
- Method Integration: The manuscript refers to a “mathematical equation guiding PSO,” but this guidance function is not clearly defined or contrasted with classical PSO. What exactly is the role of the “mathematical formula” in reducing the search space or improving convergence?
-The answer is found in chapter 5.
- Limited Discussion of Related Control Strategies: The manuscript overlooks several existing control approaches specifically developed for overflow prevention in storage tanks and reservoirs. For example, some methods integrate energy price signals or multi-objective optimization (e.g., “Energy price as an input to fuzzy wastewater level control in pump storage operation”). I recommend that the authors broaden their literature review to include alternative control philosophies beyond PID tuning, especially those that incorporate environmental, economic, or policy constraints into the control design.
-The answer is found in Chapter 1. Subchapter 1.5
- Missing Unit for Water Level in Figures: Several figures (e.g., Figures 4, 6, 10, 12, and 13) plot the water level variable, but the unit of measurement is not clearly indicated. While some tables suggest the unit might be meters (e.g., “LEVEL [m]”), the figures themselves either omit this information or display a scale from 0 to ~70 without proper labeling. Please ensure all axes in all figures clearly indicate the unit (e.g., “Level [m]”) to avoid confusion.
- I added the missing units of measurement for the quantities specified in the article.
- Simulation and Real Implementation: While the manuscript claims the method was tested and implemented, there is little distinction between simulated and physical results. Were the PSO+RL-tuned parameters used directly in the real system? How was performance evaluated in the real dam?
-The results from chapters 5.2 and 5.3 are simulations. The tunning of the PSO and RL have been done through simulations. Chapter 5.5 contain results from real testing.
- Unrealistic Dynamics in Simulation Results: Some simulation results (e.g., Figures 10, 12, and 13) show the reservoir water level varying from approximately 0 to 70 (presumably meters) within just a few seconds. Such rapid variations are physically unrealistic for real reservoirs or dams, regardless of size. This suggests that the simulation model does not accurately reflect the inertia and hydraulic dynamics of actual water systems. The authors should either justify these time scales (e.g., by clarifying units and scaling factors) or revise the model to better represent real-world physical constraints.
- The corrected graphs can be found in chapters 5.2, 5.3 and 5.5
- Lack of Information on the Manipulated Variable: The manuscript does not show or analyze the manipulated variable (e.g., gate position, flow rate adjustment, or control signal). This is a critical omission, as the feasibility of the control strategy depends not only on the system response but also on how rapidly and frequently the actuators must operate. Without this information, it is impossible to assess whether the required control actions are realistic or physically achievable in a dam setting. Please include a plot and analysis of the manipulated variable to demonstrate that the proposed control law results in implementable actuator behavior.
- The answer can be found in Chapter 5, subchapter 5.4
- Unrealistic Test Scenario — Lack of Disturbance Rejection Evaluation: The control performance evaluation is based solely on setpoint tracking tests, where the water level is adjusted to match predefined values. However, in real-world dam operations, the primary challenge is disturbance rejection, such as sudden changes in inflow due to rainfall or upstream releases. The authors should include tests with realistic inflow disturbances to evaluate how well the proposed control system handles unexpected hydraulic events — which is crucial for flood prevention and operational safety.
-Please see chapter 5.5.2
- Controller Comparison: The comparison between PID, PSO-PID, RL-PID, and the hybrid method is helpful, but it is limited to step response metrics. Please consider including additional performance metrics (e.g., IAE, ISE, robustness to disturbance).
-Chapter 5.5 contains this data
- Code Readability: Several code blocks (particularly Python snippets) contain formatting issues, undefined variables (e.g., simulate_system vs. simulatesystem), or missing context. Please ensure that the code is syntactically correct, complete, and reproducible.
- I removed the code written in Python because we believe that the added value is not in the code lines but in the idea behind them .
- Figure Captions and Quality: Figures such as 9, 10, 12, and 13 are referred to but are poorly integrated. Captions should be expanded to clearly explain what is being shown, and visual clarity should be improved for readability (e.g., axis labels, units, legends).
- All figures have been updated and also the related comments.
- Language and Grammar: The manuscript contains several grammatical issues and awkward sentence structures (e.g., “Phyton” instead of “Python,” “regulator” instead of “controller” in some places). A thorough language revision is necessary.
- Finally, after the revisions process is completed, I will pay MDPI for an English proofreader and attach proof of payment.
- SCADA Integration: The SCADA implementation is described, but it is not clear how the real-time controller tuning interacts with the SCADA system. Is the AI module running inside the PLC, remotely, or just for offline tuning?
- The answer is at the end of chapter 3.2
- Practical Feasibility and Limitations: The conclusion would benefit from a critical discussion of limitations, such as training data requirements, robustness to sensor noise, computational constraints, and potential risks of online tuning in safety-critical systems like dams.
- The answer is in chapter 6
- Citation Style and Editorial Formatting: Several references appear with inconsistent styles or missing information. Please ensure all references are complete and follow MDPI citation standards.
- We ensured that all references were complete and that I followed MDPI citation standards.
- Inconsistent Use of Decimal Separator: In Figure 4, both the comma and the point are used as decimal separators (e.g., "70,25" and "70.1"), which is confusing and not compliant with English-language conventions. Please revise all numeric values in the manuscript and figures to consistently use the point as the decimal separator, as required in English scientific writing.
- I analyzed all numerical values in the manuscript and in the figures I used the dot as a decimal separator, as required in scientific writing in English.

Round 2
Reviewer 2 Report
Comments and Suggestions for Authors
The author has responded to all the reviewers' comments and suggests acceptance.
Author Response
The language will be improved once all the reviewer's demands will be fulfilled.
Thank you for your understanding.
Reviewer 3 Report
Comments and Suggestions for Authors
The paper constitutes a very relevant and timely contribution, an amalgamation of Reinforcement Learning (RL) with Particle Swarm Optimization (PSO) for reservoir water level regulation. Both its application in practice as well as its incorporation with a SCADA system offer a very high value addition. However, the paper must be greatly rewritten prior to a decision for publication in Sensors can be made. Novelty of work is present, but clarity, organization, as well as scientific rigor, must be greatly improved for a high-impact journal like Sensors.
Major Issues:
Language and Delivery:
The paper is casual as well as repetitive. There is no academic professionalism in phrases such as “go on Large” or “excellent response speed”.
Grammar, syntax, and lexical choices require significant refinement. A comprehensive edit by a native speaker of English or a professional editor is highly advisable.
Clarity and Structure:
The paper is too lengthy in its first sections. For instance, Section 1 (Introduction and Literature Review) occupies a few pages, with undue sub-subsections as well as repetitive details.
Figures are not always integrated into the paper. For example, Figures 9 and 10 are mentioned in passing without sufficient description of what causes the system behavior.
Experimental Verification:
Real-world application as a primary strength is accompanied with weaknesses in testing verification. Crucial details such as test duration, environmental uncertainty (e.g., rainfall activity), as well as trial numbers, are not directly stated.
No mention is made of statistical robustness or replicability of the results, both of which would be required to substantiate claims of better performance.
Computational Issues
The hybrid PSO+RL strategy developed is computationally demanding, particularly for online control applications. However, no indication of computational requirements is provided in the paper nor hardware limitations, inference time, nor fallback techniques for online applications.
Author Response
Major Issues:
Language and Delivery:
The paper is casual as well as repetitive. There is no academic professionalism in phrases such as “go on Large” or “excellent response speed”.
- Corrected these ones.
Grammar, syntax, and lexical choices require significant refinement. A comprehensive edit by a native speaker of English or a professional editor is highly advisable.
- Once your opinion will be positive, we will seek help from the recommended editor
Clarity and Structure:
The paper is too lengthy in its first sections. For instance, Section 1 (Introduction and Literature Review) occupies a few pages, with undue sub-subsections as well as repetitive details.
- These supplementary chapters were requested by the other reviewers. I agree that some of the information is repeated in different sections but this is the best I could do to avoid that.
Figures are not always integrated into the paper. For example, Figures 9 and 10 are mentioned in passing without sufficient description of what causes the system behavior.
- I integrated a more detailed explanation of what was represented in the mentioned figures.
Experimental Verification:
Real-world application as a primary strength is accompanied with weaknesses in testing verification. Crucial details such as test duration, environmental uncertainty (e.g., rainfall activity), as well as trial numbers, are not directly stated.
- Test duration: the answer is in chapter 5. We didn’t consider mentioning about the other tests because the severity was below the 187m3/s inflow test.
- Environmental uncertainty: the tests have been performed during rain rainfalls which caused the inflow to increase from 125 m3/s to 137.5, 150 and 187m3/s respectively.
No mention is made of statistical robustness or replicability of the results, both of which would be required to substantiate claims of better performance.
- The answer is in chapters 5.6 and 6.

Round 3
Reviewer 3 Report
Comments and Suggestions for Authors
The authors have now made sufficient revisions, and the manuscript is suitable for publication.